# Deep-Ocular: Improved Transfer Learning Architecture Using Self-Attention and Dense Layers for Recognition of Ocular Diseases

**DOI:** 10.3390/diagnostics13203165

**Published:** 2023-10-10

**Authors:** Qaisar Abbas, Mubarak Albathan, Abdullah Altameem, Riyad Saleh Almakki, Ayyaz Hussain

**Affiliations:** 1College of Computer and Information Sciences, Imam Mohammad Ibn Saud Islamic University (IMSIU), Riyadh 11432, Saudi Arabia; mmalbathan@imamu.edu.sa (M.A.); altameem@imamu.edu.sa (A.A.); ralmakki@imamu.edu.sa (R.S.A.); 2Department of Computer Science, Quaid-i-Azam University, Islamabad 44000, Pakistan; ayyaz.hussain@qau.edu.pk

**Keywords:** computer-aided diagnosis, transfer learning, multi-label classification, ocular diseases, diabetic retinopathy, glaucoma, cataract, XgBoost, AlexNet, deep learning

## Abstract

It is difficult for clinicians or less-experienced ophthalmologists to detect early eye-related diseases. By hand, eye disease diagnosis is labor-intensive, prone to mistakes, and challenging because of the variety of ocular diseases such as glaucoma (GA), diabetic retinopathy (DR), cataract (CT), and normal eye-related diseases (NL). An automated ocular disease detection system with computer-aided diagnosis (CAD) tools is required to recognize eye-related diseases. Nowadays, deep learning (DL) algorithms enhance the classification results of retinograph images. To address these issues, we developed an intelligent detection system based on retinal fundus images. To create this system, we used ODIR and RFMiD datasets, which included various retinographics of distinct classes of the fundus, using cutting-edge image classification algorithms like ensemble-based transfer learning. In this paper, we suggest a three-step hybrid ensemble model that combines a classifier, a feature extractor, and a feature selector. The original image features are first extracted using a pre-trained AlexNet model with an enhanced structure. The improved AlexNet (iAlexNet) architecture with attention and dense layers offers enhanced feature extraction, task adaptability, interpretability, and potential accuracy benefits compared to other transfer learning architectures, making it particularly suited for tasks like retinograph classification. The extracted features are then selected using the ReliefF method, and then the most crucial elements are chosen to minimize the feature dimension. Finally, an XgBoost classifier offers classification outcomes based on the desired features. These classifications represent different ocular illnesses. We utilized data augmentation techniques to control class imbalance issues. The deep-ocular model, based mainly on the AlexNet-ReliefF-XgBoost model, achieves an accuracy of 95.13%. The results indicate the proposed ensemble model can assist dermatologists in making early decisions for the diagnosing and screening of eye-related diseases.

## 1. Introduction

A key challenge in healthcare is identifying eye disease using fundus images [1]. Any ailment or disorder that impairs the eye’s ability to function normally or negatively affects the eye’s visual acuity is referred to as an ocular disease [2]. Almost everyone experiences visual issues at some point in their lives. Some require the care of a specialist, while others are minor that do not appear on healthcare claims or may be handled at home [3]. Globally, fundus problems are the main reason why people go blind. The most prevalent ocular diseases (AMDs) include age-related macular degeneration (AMD), cataracts, glaucoma, and diabetic retinopathy (DR). By 2030, more than 400 million people will have DR, per related research [4]. These eye conditions are becoming a significant worldwide health issue.

Most importantly, the ophthalmic condition is fatal and may leave patients permanently blind. In clinical settings, the early detection of these illnesses can prevent visual damage. The number of ophthalmologists and the number of patients, however, are significantly out of proportion. Additionally, manually evaluating the fundus takes a lot of time and is highly dependent on the expertise of the ophthalmologists. This makes thorough fundus screening more challenging. Therefore, automated computer-aided diagnostic methods are essential for identifying eye problems. This is a typical opinion [4].

Globally, there is a vast range of prevalence rates of eye disorders based on age, gender, employment, lifestyle, economic status, cleanliness, habits, and conventions. According to an article of research that studied individuals in tropical and temperate countries, unavoidable eye infections are more common in low-altitude populations because of variables including dust, humidity, sunshine, and other natural elements [5]. Additionally, eye disorders present differently in communities in developing and industrialized nations. There are actual rates of ocular morbidity in many developing countries, notably in Asia, that are underdiagnosed and neglected [6]. Globally, 285 million individuals are estimated to have visual impairments, of which 246 million have poor eyesight and 39 million are blind [7]. The World Health Organization (WHO) estimates that 2.2 billion individuals worldwide have a close-up or distant vision issue [8]. Estimates suggest that half of these situations might have been avoided or resolved [9]. The primary causes of visual impairment include cataracts, age-related macular degeneration, glaucoma, diabetic retinopathy, corneal opacity, trachoma, hypertension, and other conditions [10]. A microscopic study was conducted on the prevalence of blindness and visual impairment in Bangladesh. Most of the population lives in rural areas. Over 80% of people who live in cities nowadays need medical attention, yet few ophthalmology services are available. Although more businesses provide services for blindness, the incidence is still low [11].

In practice, there may be a higher risk of vision loss due to the increased prevalence of non-communicable diseases, including diabetes and smoking [12]. Many impoverished people living in slums have poor mental and physical health [13]. Given this, offering these people affordable or accessible complete eye care treatments is critical [14]. Medical image analysis is using more and more deep-learning-based techniques [15]. Deep-learning-based models have been effective in various tasks, including illness identification [16]. One of the most essential steps in minimizing an ophthalmologist’s workload is the automated diagnosis of diseases. Without human interaction, illnesses may be detected using deep learning and computer vision technology. Only a small number of these studies were able to fully diagnose more than one eye illness [17], even though many of them produced encouraging results. To accurately detect diverse eye conditions, more study is required to examine the many elements of fundus imaging [18]. This study suggests a deep learning (DL) technique for recognizing eye illnesses. The most common eye-related diseases, as shown in Figure 1, are briefly explained below.

(a)Diabetic retinopathy: The persistently elevated blood sugar levels caused by diabetes can damage the microscopic blood vessels (capillaries) that transport oxygen and nutrients to the retina. Diabetic retinopathy affects approximately one-third of people with diabetes over 50.(b)Cataract: A cataract is the clouding of the lens of the eye. Cataracts can eventually contribute to blindness if left untreated. People with diabetes are more likely to develop cataracts and experience vision loss at a younger age than those without the condition.(c)Glaucoma is a group of conditions that can cause an optic nerve injury. The optic nerve transmits retinal signals for processing to the brain. Glaucoma is frequently (but not always) caused by increased intraocular pressure. Diabetes substantially increases the risk of glaucoma in comparison to the general population. Open-angle glaucoma (also known as “the sneak thief of sight”) and angle-closure glaucoma (which develops abruptly and is a medical emergency) are the two most common varieties.

Multilabel categorization has been used as a different strategy [19]. Ocular disease datasets [20,21] could be more balanced. This imbalance makes it difficult to accurately identify or classify sickness or even a standard retinograph image. This method is not recommended for broad classification problems due to its low accuracy. The classification of ocular illnesses is the goal of this effort. It is not suggested to categorize any disease using the dataset used in this study since it was very unbalanced. This mismatch caused a lot of variability throughout training, which left a lot to be desired. The strategy we used to address the issue was balancing the images of the classes. We balanced the ranks by taking the same number of pictures from each category and feeding them into a pretrained AlexNet model rather than utilizing all the images and categorizing all the illnesses simultaneously. As a result, our study initially balanced the dataset by training the classes on the pretrained AlexNet architecture with the same amount of data for each category. By selecting an equal number of images for each type, we first loaded the dataset and the associated image into the dataset. The AlexNet model used the transfer learning (TL) approach in this research. The accuracy of each class rose once we correctly balanced the dataset.

We developed a deep-ocular detection system based on retinal fundus images. The advantage of improving the AlexNet (iAlexNet) architecture by incorporating attention and dense layers, compared to other transfer learning architectures, lies in its ability to enhance feature extraction, adaptability, and interpretability for specific tasks. To create this system, we used ODIR and RFMiD datasets, which included various retinographics of distinct classes of the fundus, using cutting-edge image classification algorithms like ensemble-based transfer learning. These models, however, only employ one deep learning network to identify eye-related diseases; hence, the efficacy of this method still has to be enhanced. In this paper, we suggest a three-step hybrid ensemble model that combines a classifier, a feature extractor, and a feature selector. The original image features are first extracted using a pre-trained AlexNet model with an enhanced structure to overcome the overfitting and gradient vanishing problems. The extracted features are then selected using the ReliefF method, and then the most crucial elements are chosen to minimize the feature dimension. Finally, an XgBoost classifier offers classification outcomes based on the desired features.

### 1.1. Research Significance

(1)A deep-ocular detection system is developed in this paper to address the issues of the multiclass recognition of eye-related diseases based on ensemble model by combing iAlexNet, ReliefF, and XgBoost.(2)A preprocessing step is performed by integrating CLAHE and Ben Gram contrast adjustment techniques. A data augmentation step is also applied to adjust class imbalance and overfitting problems.(3)A new model iAlexNet, developed using attention and dense layers, allows the network to focus on important regions within an image. This is particularly beneficial for tasks where certain regions carry critical information (e.g., diseases in medical images). Attention mechanisms enable the model to assign varying levels of importance to different parts of an image, leading to improved feature extraction.(4)The introduction of additional dense layers further refines the features extracted by the convolutional layers in the iAlexNet model. Dense layers capture high-level abstractions and relationships among features, enabling the model to learn more complex patterns and representations.(5)The improved iAlexNet architecture allows for the fine-tuning of specific layers, making it adaptable to a wide range of tasks. By unfreezing selected layers during training, the model can specialize its learned representations to match the requirements of the target task. This adaptability is especially valuable when working with limited labeled data.

### 1.2. Paper Organization

The remaining sections of this paper are divided into five sections. Section 2 contains related works. It discusses the current limitations and emphasizes the primary directions and solutions incorporated into the proposed system to address the current deficiencies. Section 3 describes in detail the phases and techniques the proposed deep-ocular framework utilizes. Section 4 describes and presents the results of the various investigations conducted. Section 5 introduces the discussion and provides a comparative, analytical examination of the proposed system and other cutting-edge techniques. Section 6 summarizes the paper’s findings and highlights future research directions.

## 2. Related Work

Ophthalmology has witnessed significant advancements with the advent of artificial intelligence (AI) and deep learning [22]. This literature review sheds light on pivotal studies focusing on these advancements, emphasizing the range of diseases tackled, methodologies employed, and the results achieved. The state-of-the-art studies are described in Table 1.

In the early days of applying AI to ophthalmology, ref. [23] made strides by proposing a method for optic disc (OD) localization using convolutional neural networks (CNNs). Their innovative data preparation and two-stage training process adeptly addressed the class-imbalance issue, demonstrating a remarkable detection rate of 99.11%. Unique to this approach was the replacement of the less informative blue channel with segmented vasculature maps, providing richer context for the CNN structures. Continuing on the trail of diabetic retinopathy (DR), ref. [24] showcased a hybrid DL network to detect and grade DR severity. With the utilization of ResNet50, they achieved an accuracy of up to 96% on the IDRiD dataset. The potential of AI in improving clinical outcomes became even more evident with [25], where a single network was able to predict the severity of DR with high sensitivity and specificity.

Expanding the AI spectrum, ref. [26] addressed the global challenge of cataracts, a predominant cause of blindness. They proposed an efficient cataract detection model using the VGG19 network, which minimized the need for manual retinal feature extraction. This research marked a critical transition towards automating ophthalmological diagnosis. Ref. [27] further highlighted the inherent limitations of manual diagnoses due to constrained medical resources. They released the comprehensive OIA-ODIR dataset and benchmarked the performance of nine prominent CNNs, suggesting the importance of feature fusion for multi-disease classification.

Recognizing the importance of timely examinations, ref. [28] explored the application of AI in Diabetic Eye Disease (DED). They presented a novel model for multi-class DED classification, achieving an accuracy of 81.33%. Their work established the potential of optimized deep learning architectures, especially in scenarios with limited data availability. Moving ahead, refs. [29,30] concentrated on diagnosing various eye diseases using advanced deep learning algorithms, achieving significant accuracies and emphasizing the efficiency of automated systems.

The global shortage of ophthalmologists is a growing concern, and ref. [31] delineated the promise of AI in addressing this. The study demonstrated the efficacy of deep learning in classifying eye diseases even with images of minimal resolution. Following this trend, ref. [32] detailed a unique approach for detecting glaucoma by employing a CNN based on the Resnet152 model, highlighting the enhanced accuracy potential of machine learning.

Recent advancements have also underscored the vast database potential, as seen with [33], in which the authors developed an algorithm using a vast collection of AS-OCT images. Furthermore, ref. [34] introduced EyeDeep-Net, a multi-layer neural network designed to efficiently diagnose various eye problems, setting a benchmark in the field. Of note, ref. [35] provided a nuanced approach, deploying a DCNN optimized with a whale optimization algorithm. With a significant accuracy leap of 8.1%, this model effectively detected and classified a range of eye diseases, from cataracts to AMD. Concluding this review, ref. [36] brought forth a groundbreaking patient-level multi-label ocular disease classification model, the DCNet. This model, with its three integrated components, presented a holistic approach to patient diagnostics, further accentuating the role of AI in revolutionizing ophthalmological diagnostics.

## 3. Materials and Methods

The overall steps are represented in Algorithm 1 and a systematic flow diagram of the proposed system is presented in Figure 2. The deep-ocular system consisted of different phases such as preprocessing, data augmentation, an improved AlexNet architecture, feature selection and feature classification. To develop this deep-ocular system, we integrated the below steps, and briefly describe them in the subsequent paragraph.

The accuracy and efficiency problems in the multiclass diagnosis of glaucoma (GA), diabetic retinopathy (DR), cataract (CT), and normal (NL) eye-related diseases were resolved with the use of an advanced hybrid model containing feature extraction, feature selection, and classification components. This work suggests a three-step hybrid ensemble model including a feature extractor, feature selector, and classifier to address this issue. The retinograph image was preprocessed, and features were first extracted using an upgraded version of iAlexNet as a feature extractor. The ReliefF method then assigned a priority score to each extracted feature. The XgBoost classifier received the first n features as the input after determining the optimal number of input features (n) using the trial-and-error approach. The XgBoost classifier then output the classification results based on the n chosen features, i.e., the DR, CT, GL, and NL classes of eye-related disease.
**Algorithm 1:** Outline for using the improved iAlexNet architecture for feature extraction, applying the Relief algorithm for feature selection, and then classifying the information features using XgBoost classifierStep 1Let X be the dataset of eye images. Let Y be the corresponding class labels for the images. AlexNet(), $f_{AlexNet}(x)$ represents the feature extraction process using the modified AlexNet. Let F be the set of selected features after applying the Relief algorithm. Let D be the distribution of the data. Let H be the AdaBoost to predict the retinograph class (DR, CT, GL, and NL).Step 2Load pretrained AlexNet model: load a pretrained AlexNet model: $m_{AlexNet}$. Step 3Improved AlexNet architecture: load AlexNet base model with ImageNet weights, base_model, incorporate additional convolutional and attention layers, and add dense layers to refine features: FeaturesEnhancement(Self, FeatureMaps). Feature extraction: $f_{AlexNet}\left(x\right)=m-iAlexNet(m_{AlexNet},x)$
Step 4Fine-tuning and early stopping: unfreeze selected top layers of the model, recompile model with reduced learning rate: {model.compile(optimizer = Adam(lr = 0.0001), loss = ‘categorical_crossentropy’, metrics = [‘accuracy’])}.Implement early stopping to prevent overfitting: {early_stopping = EarlyStopping(patience = 3, restore_best_weights = True)}.Step 5Apply relief algorithm for feature selection: F = Relief(X,Y).Step 6Train gradient boosting classifier on selected features: gb_classifier.fit(X_train_selected, y} {{train}}).Step 7Predict class labels using the classifier: {y_pred = gb_classifier.predict(X_test_selected)}.Calculate accuracy by comparing predicted and actual labels: {accuracy = accuracy_score(y}_{{test}},{y_pred)}.Step 8Output: display calculated accuracy and predicted class label for the new retinograph.

### 3.1. Data Acquisition

The performance of the proposed hybrid system was evaluated based on two publicly available datasets, i.e., the retinal fundus multi-disease image dataset (RFMiD) [37] and ODIR (Ocular Disease Intelligent Recognition) [38].

A dataset called ODIR (Ocular Disease Intelligent Recognition) was employed in this work. This dataset is one of the most extensive public resources on Kaggle for identifying eye illnesses. Eight classifications of ocular diseases are usually presented. However, we selected only diabetes, cataracts, glaucoma, and normal eye-related diseases. All of the photographs for this project were scaled to 224 × 224, as described in Table 2.

The retinal fundus multi-disease image dataset (RFMiD) focuses primarily on diabetic retinopathy, glaucoma, age-related macular degeneration, and a few additional prevalent pathologies. This paper aimed to unite the medical-image-analysis community in order to develop methods for the automatic classification of both common and rare ocular diseases. The retinal fundus multi-disease image dataset (RFMiD) consists of 3200 fundus images captured with three distinct fundus cameras and annotated with 46 conditions by two senior retinal experts. According to the authors’ knowledge, RFMiD is the only publicly accessible dataset comprising such a wide variety of frequently seen diseases in clinical settings. This challenge encouraged the development of generalizable models for screening the retina, in contrast to previous efforts that focused on the detection of specific diseases.

### 3.2. Preprocessing and Data Augmentations

The primary phases of the preprocessing stage were image resizing and data enhancement. These steps were performed on each retinograph image. Image Resizing: Every image was scaled to 224 × 224 pixels. We performed CLAHE and Bengram contrast enhancement techniques on every input image of size 224 × 224. This step increased the contrast of the pixels so that classification accuracy would be increased. Although using the actual dimensions could be advantageous for learning, it is necessary to resize the input images to conserve memory and reduce training time.

Each image in the employed dataset contains multiple eye-related diseases from the retinograph images. Nonetheless, this dataset needs to be more balanced since the number of standard samples may vastly outnumber those of eye-related disease classes. In the training dataset, the number of standard images (i.e., negative samples) containing no eye diseases was 4000. In contrast, the number of images containing specific pathologies, such as glaucoma, diabetic retinopathy, and cataract, was less than 400. Therefore, data augmentation techniques should increase the number of images containing rare diseases and reduce the positive–negative class imbalance. Various transformation techniques can be applied to the image dataset to expand the dataset, address class imbalances, and prevent overfitting. We used basic, safe, and manageable augmentation techniques in this study to preserve labels and encourage the model to learn more general features.

Its propensity to maintain the label after transformation serves as a gauge of a data augmentation method’s safety. To ensure that each label in the dataset occurs at least 100 times, we augmented images of rare diseases using horizontal and vertical flipping, rotation, luminance modification, saturation modification, and hue modification. Table 3 details the employed augmentation techniques and their respective parameter values. This step is visually displayed in Figure 3.

### 3.3. Model Architecture

The improved AlexNet (iAlexNet) architecture offers the above advantages. Other transfer-learning architectures like ResNet, VGG, and Inception can also excel in different scenarios based on their unique characteristics. It is recommended to experiment with multiple architectures and analyze their performance in the context of the specific task at hand. All of the retinograph images were sent into AlexNet for feature extraction, and the network’s output from a particular layer was considered for classifying eye-related diseases. Transfer learning allows standard features from trained convolutional neural networks to be used as input classifiers for imaging tasks, since most computer-aided diagnosis (CAD) systems and other medical-image interpretation systems cannot train convolutional neural networks from scratch. The ReliefF method was used to rank the characteristics that were retrieved in the first phase according to their relevance. The initial features most crucial for categorization were then chosen through trial and error. The XgBoost model is described in Section 3. The XgBoost model classifies the filtered features after receiving the previously selected features as input, producing final classification results.

#### 3.3.1. AlexNet Model for Feature Extraction

AlexNet is a transfer learning (TL) algorithm that can significantly increase computing performance using two GPUs for calculations [23]. Since AlexNet is a massive network with 60 million parameters and 650,000 neurons, it needs a lot of labeled examples to train [24], which the labeled DR, CAT, GL, and NL image resources cannot provide. When insufficiently labeled examples are available, transfer learning is a practical and popular technique for training deep neural networks. Utilizing every parameter in a pre-trained network as an initialization step might make use of characteristics discovered via the use of enormous datasets. The generated parameters from using these layers, primarily utilized for feature extraction, can aid in training convergence. Furthermore, transfer learning may be implemented on regular personal computers, whereas deep network training requires a high-performance GPU and CPU.

Figure 4 shows the improved AlexNet pretrained model used to recognize DR, CAT, GL and NL ocular diseases. The basic AlexNet is composed of eight layers, categorized into convolutional, max-pooling, and fully connected layers. The first five layers consist of convolutional operations, followed by some layers with max-pooling operations. The last three layers are fully connected. The network architecture is organized in a way that involves parallel processing across two GPUs, with the exception of the final layer. To develop this improved AlexNet architecture, we added the self-attention and dense layers.

We can represent the improved iAlexNet architecture as follows:○Convolutional layer (1): convolution operation with learnable filters and activation function.○Max-pooling layer (1): downsampling operation to reduce spatial dimensions.○Convolutional Layer (2): another convolution operation with activation.○Max-pooling layer (2): additional downsampling.○Convolutional layer (3): third convolution operation with activation.○Convolutional layer (4): fourth convolution operation with activation.○Convolutional layer (5): fifth convolution operation with activation.○Max-pooling layer (3): further spatial downsampling.○Flatten layer: flattening the feature maps to a vector for input to fully connected layers.○Fully connected layer (1): neurons fully connected to the flattened feature vector.○Fully connected layer (2): second fully connected layer.○Fully connected layer (3): final fully connected layer (output layer).○Convolutional layer (6): fifth convolution operation with activation.○One self-attention layer.○Four convolutional layers (10): fifth convolution operation with activation.○One max-pool layer.○One global average pool layer.○Two dense layers.

The last two layers of the proposed model, a fully connected layer with 1000 neurons and a softmax layer, were replaced by our layers, which were two fully combined layers with ten and three nodes (representing the three types of categories: DR, CAT, GL, and NL). (These layers are shown in Figure 4) The layers improved the original AlexNet architecture. The remainder of the original model’s parameters were kept and used as initialization. The pre-trained and transferred networks were then separated, leaving the whole structure in two pieces. The extracted features were effective for classification, and the parameters in the pre-trained network had previously been trained on ImageNet with millions of retinographics. Some settings might have only needed to be slightly changed to modify them for the new retinographics. For training on a limited dataset, the parameters in a transferred network only represent a tiny portion of the complete network.

An approach to incorporate self-attention into images involves utilizing a CNN for extracting feature maps from the initial image. This step is described in Algorithm 2. Following this, self-attention is employed on these feature maps. The self-attention mechanism can be employed in different manners on the feature maps, including spatial self-attention or channel self-attention.

Spatial self-attention encompasses the calculation of attention weights based on spatial correlations among various regions within the feature maps. This enables the model to concentrate selectively on distinct areas of the image, considering their spatial interrelationships. This proves advantageous for tasks like object recognition, where the spatial arrangement of object components holds significance.

Channel self-attention, on the other hand, computes attention weights by assessing the relationships among different channels within the feature maps. This empowers the model to focus discriminately on particular channels based on their pertinence to the given task. Channel self-attention finds utility in tasks such as image classification, where specific channels might carry greater relevance for certain classes compared to others.

Through a fusion of spatial and channel self-attention, it becomes feasible to capture both local and global dependencies present in image features. This approach facilitates targeted emphasis on the most pertinent image areas for a specific task. Empirical evidence has demonstrated that self-attention yields performance enhancements across various computer vision tasks, including image classification.
**Algorithm 2:** FeaturesEnhancement(FeatureMaps): Enhances the features of the given feature maps using a self-attention mechanism
Input FeatureMaps: The feature maps are extracted from an AlexNet CNN model.
Output: Enhance the features of the given feature maps using a self-attention mechanism.Step 1Extract feature maps using the CNN model: FeatureMaps = Extract-Maps().Step 2Compute spatial attention: ComputeSpatialAttention(FeatureMaps): begin.
Repeat for position in FeatureMaps:
 AttentionScores = SpatialAttentionScores(position)
 WeightedSumFeatures = WeightedSumFeatures(FeatureMaps, AttentionScores, position)
[End for Loop]
SpatialAttentionMaps SpatialAttention(FeatureMaps)End functionStep 3Compute channel attention: ChannelAttention(FeatureMaps): begin.
Repeat for channel in FeatureMaps:
 AttentionScores = ChannelAttentionScores(channel)
 WeightedChannelFeatures = WeightedChannelFeatures(FeatureMaps, AttentionScores, channel)
 ChannelAttentionFeatureMaps = ChannelAttention(FeatureMaps)
[End for loop][end function]Step 4CombineAttention(SpatialAttentionFeatureMaps, ChannelAttentionFeatureMaps)Step 5EnhancedFeatureMaps = ElementWiseMultiply(SpatialAttentionFeatureMap, ChannelAttentionFeatureMap)Step 6EnhancedFeatureMaps = Combine (SpatialAttentionFeatureMaps, ChannelAttentionFeatureMaps)
[End of Algorithm]

#### 3.3.2. Feature Selection using Relief

ReliefF, a dimension reduction technique that Kira and Rendell developed, can help reduce computational complexity, expedite model training, and remove unused characteristics from a data set. Algorithm 3 shows the steps to select features. The ReliefF model can be enhanced by increasing the dataset’s noise resistance and adapting it to multi-class situations by disregarding missing data [24]. ReliefF seeks to identify the correlations and regularities in the dataset’s properties.
(1)$$W\left(F_{l}\right)=W\left(F_{l}\right)-\sum_{j=1}^{k}\frac{diff\left(F_{l},R_{i},H_{i}\right)}{m.k}+\;\sum_{classs\left(R_{i}\right)}\left[\frac{P\left(c\right)}{1-p\left(class\left(R_{i}\right)\right)}\sum_{j=1}^{k}diff\left(F_{l},R_{i},M_{i}\left(c\right)\right)\right]/M_{i}$$

The fundamental actions of ReliefF are given in pseudo-code format in Algorithm 3. In this study, the ten retrieved characteristics were sorted according to relevance using the ReliefF method. The training set’s feature data, not the test sets, were the ones that were utilized. Following extensive testing, we concluded that, to obtain the most significant classification speed and accuracy, just a small number of the most crucial attributes needed to be employed. The following sections will provide further examples.
**Algorithm 3:** Features selection using ReliefF methodInput: *M* learning instances with *L* features and *C* classesOutput: the vector $w=W\left[f_{1},f_{2},f_{3},\ldots ,f_{l}\right]$
Step 1[Initialized] Set all weights W (f_l) = 0, where 1,2,3…,lStep 2for i = 1 to m doStep 3Randomly select an instance $R_{i}$
Step 4Find k nearest histograms $h_{j}$ of $R_{i}$
Step 5Repeat for each class c≠ class ($R_{i})$ doStep 6From class c, find k nearest misses $m_{j}$(c) of ${\;R}_{i}$

[End for loop]Step 7For l = 1 to L, doUpdate W(F) by using Equation (1)
[end for loop]
[end for loop]Step 8[end]

#### 3.3.3. XgBoost Classifier

Based on statistical learning theory, the author created the supervised learning technique known as XgBoost [24]. The dataset was split into training and test sets before XgBoost executed the learning procedure. Currently, XgBoost is widely used for classification tasks in various areas, including text classification, bioinformatics, handwritten-character recognition, and facial recognition. XgBoost splits the initial classification issue into multi-classification problems. These steps are mentioned in Algorithm 4. Consequently, when used for multi-classification, the complexity and difficulty of training rise in steps with the number of sample categories. For XgBoost, reducing computation and computational complexity is a known issue that necessitates new research solutions [24]. We suggest using the ReliefF technique to address this issue and make the sample data less dimensional. Gradient tree boosting is one of the most effective and widely applied machine learning (ML) techniques. It excels in various applications and is a highly competent ML technique. Consequently, it has been demonstrated that it provides state-of-the-art results for various standard classification, regression, and ranking problems. We discuss the extreme Gradient Boosting algorithm below.
**Algorithm 4:** Extreme Gradient Decision-Boosting Machine (xgBoost) algorithmInput: Extracted feature data $x=(x_{1},\;x_{2},\ldots ,\;x_{n})$ with labels y and test data $x_{test}$
Output: Class labels (glaucoma (GA), diabetic retinopathy (DR), cataract (CT), and normal (NL)).Process:Step 1. First, initialize tree as a constant: $y_{i}^{t}=f_{0}=0$ for optimization and parameter defined for the classifier.Step 2. Construct XgBoost tree by minimizing the lost function using Equation (2).The XgBoost() classifier is perfectly trained on the feature samples $x=(x_{1},\;x_{2},\ldots ,\;x_{n})$.Step 3. Repeat Step 2 until the model reaches the stop condition.Step 4. Test samples $y_{i}^{t}$ are assigned the class label using the decision function of Equation (2).
(2)$$f_{t}\left(x_{i}\right)=argmin\;L\left(t\right)=\arg\min L\;(y,\;y+f\;(x))$$

The parameter settings are as follows for the XgBoost classifier: a learning rate of 0.3, max_depth of 3, gamma of 4, minchildweight of 1, subsample of 1, samplingmethod of “uniform”, lambda of 1, alpha of 0, objective of binary: logistic, nestimators of 50, and nthread of 16.

### 3.4. Fine-Tune Network

Fine-tuning is a technique used in transfer learning (TL), a process in which a pre-trained neural network is adapted to a new task or dataset. When a neural network is trained on a large dataset for a specific task, it learns to recognize various features and patterns within the data. Fine-tuning allows us to leverage this learned knowledge and adapt the network to perform well on a related but different task. Fine-tuning strikes a balance between retaining general features learned from the original task and adapting to the specifics of the new task. It is particularly useful when we have limited data for the new task, as it helps prevent overfitting by leveraging the knowledge contained in the pre-trained layers. By freezing and updating layers strategically, fine-tuning allows the network to adjust its representations to match the new data distribution, ultimately leading to improved performance on the new task.

In the context of deep learning, the term “fine-tuning” generally refers to two main steps: freezing and updating layers.

In the first step of fine-tuning, we freeze a portion of the pre-trained network’s layers. These frozen layers are kept fixed and unchanged during the training process. By doing so, we ensure that the network retains the knowledge it acquired from its original task, and we focus the training process on adapting the later layers to the new task.After freezing the initial layers, we introduce new layers on top of the frozen ones. These new layers are initialized randomly and will be trained from scratch. The network is then trained on the new dataset using the new layers while keeping the earlier layers frozen. This step allows the model to learn task-specific features from the new dataset without drastically altering the knowledge it gained from the original training.

Fine-tuning is a powerful technique, but it requires the careful consideration of hyperparameters, and a good understanding of the domain and the specific tasks involved. Hence, we performed the following fine-tune steps to the network.

Freezing the layers of the pre-trained AlexNet model by setting layer.trainable = False to prevent them from being updated during the initial training.Compiling the model with a lower learning rate and training it using the new dataset.Unfreezing the top few layers of the model (model.layers [−6:]) to allow them to be fine-tuned.Recompiling the model with a lower learning rate and continuing training using the same data generator and new dataset. The EarlyStopping callback was implemented to prevent overfitting.

## 4. Experimental Results

### 4.1. Experimental Setup

All deep learning (DL) models involved in the study were pre-trained on ImageNet, and their training and testing were conducted using the publicly available datasets. This framework was compiled with CUDA 8.0 and CUDNN 5.1 to enhance performance and efficiency. The experiments were executed on a high-powered workstation with three Intel CPUs, and a substantial 64 GB of memory. Ensuring smooth and seamless operations, the workstation operated on the Windows 2022 Server operating system. This robust hardware and software setup facilitated the comprehensive training and testing of the deep networks, ensuring accurate and reliable results in the evaluation process.

In evaluating the performance of multi-label eye-disease classification networks, we employed five assessment metrics, i.e., accuracy (AC), sensitivity (SE), specificity (SP), the F1-score (F1), and the AUC, and their average, termed as the Final-score. The F1, representing the harmonic mean of precision and recall, attains high values only when both precision and recall are elevated. Given that Kappa and F1 focus on a single threshold, and considering the probabilistic output of classification networks, the AUC (area under the ROC curve) was utilized to effectively assess multiple thresholds. These five metrics were computed using the sklearn package, ensuring a thorough and efficient evaluation process. 

Python’s pre-trained iAlexNet and transfer learning (TL) techniques were used to refine the model. Our proposed system accuracy and loss for both training and validation after 300 epochs are shown in Figure 5.

The specific fine-tuning technique was the addition of a fully connected layer with 10 neurons between layers 22 and 23 (between fc8 and drop7); the outputs of this layer were features retrieved for further selection. The original FC8 layer was made up of 1000 neurons, which can classify 1000 different sorts of images. The number of neurons in this layer was fixed at 4 because our classification findings only contained four categories: glaucoma (GA), diabetic retinopathy (DR), cataract (CT), and normal (NL).

After the model structure was modified to better meet the classification aims of this study, the model was trained to fine-tune the weights using the training set. All of the data were fed into the model after it had been trained to extract 10 characteristics from a total of 1743 images, including from the test set (521 images) and the training set (1222 images). By using stochastic gradient descent with momentum (SGDM), the transferred AlexNet was trained. Table 4 lists the training settings for AlexNet.

### 4.2. Performance Evaluation

The CNN-based architecture AlexNet was employed in our experiment to evaluate model performance. The enhanced retinograph image was initially collected from the dataset in the data sequencing and splitting phase, and the data were then transformed into train labels and target labels. The training–test split technique from the scikit-learn package was applied. A total of 70% of the data were used for training and 30% for testing, dividing the total into a 70:30 ratio. In this part, both models’ performance metrics and prediction skills are shown. The following are some of the outcomes for each class: During the process of training our models using the training data, we determined how much our models are learning from the training dataset. The primary objective of training precision is to extract the hyperparameters and determine whether our models suffer from overfitting or underfitting. When we completed training our models with the training dataset and validated their performance in the validation dataset, only then could we proceed to the test accuracy, which is the final accuracy of our models. When we refer to accuracy in this paper, we are referring to test accuracy. Sometimes, veracity alone is not sufficient. We could not conclude that our model was very accurate based solely on its accuracy because, in this paper, we had to correctly classify both diseases and no diseases. Those who have a disease were termed “positive” in terms of deep learning, whereas those who do not have a disease were termed “negative.” Precision provides a distinct picture of how many disease patients are correctly identified within the entire dataset. Sometimes, even precision is insufficient. For instance, if the dataset is highly biased towards one target, recall provides the number of correctly classified true positives, i.e., those individuals who are truly diseased and for whom our model made a disease prediction. This recall was the most essential metric for our research project because if we have a low recall, our model may incorrectly predict a diseased individual.

The F1 score is often referred to as the harmonic mean of precision and recall. If someone asserts that precision and recall are of equal importance, then he or she should focus on the F1 score. In this research, the F1 score was our second-highest priority after the recall. In addition, confusion matrices were utilized in this study’s model evaluation. The confusion matrix is a common error matrix used to evaluate the efficacy of supervised learning algorithms. Each column in a confusion matrix represents the expected category, and the sum of each column represents the total number of expected data points for that category. The total quantity of data in each row indicates the number of instances of that category in the data, and each row displays the data’s actual attribution category.

The accuracy of the classification results may reach 95.33% when the top 20 significant features are used as the XgBoost classifier’s input, and is the greatest value when compared to other numbers of input features. Even though accuracy can remain constant when only the first seven characteristics are entered, a larger number of features results in a longer training period for the model. The top five characteristics provided by the iAlexNet and ReliefF algorithms were, therefore, found to be the best ones for this application. Table 5 shows the different TL models to classify ocular diseases by using different epochs and combination of various architectures.

We conducted numerous independent repeated tests to assess the strength of the proposed model in terms of accuracy, specificity, and sensitivity compared with several existing models for the accuracy comparison in the following sections. The best feature number n was chosen for each separate repeat experiment in accordance with the unique experimental findings, and the value of n was necessarily set to 300. The model’s generality will not have been impacted, though, because we only needed to know the value of the ideal feature number n once throughout the model training process.

Four models were compared in the experimental analysis that follows and is addressed in Section 4. Improved AlexNet, ReliefF, and XgBoost were used in the suggested model compared to other state-of-the-art deep learning models. Those models were also constructed to test the efficacy of the model components suggested in this paper. In Figure 6, the model accuracy and loss for both training and validation after 300 epochs are shown. The experimental findings in Table 6 demonstrate that all models had acceptable accuracy. The proposed model improved on the original AlexNet’s structure in comparison to the model. According to the results, the two models’ accuracies were comparable, but the improved model (improved AlexNet) may significantly help the overall model perform better because if model is built using one model rather than another, it must classify 1000 features, which significantly lengthens the training time without clearly improving accuracy. A visual result of the proposed system is presented in Figure 6.

The proposed model created an XgBoost model to categorize the retrieved features after using AlexNet to extract the original image’s features. The accuracy of the model was found to be 95.642 ± 0.398% when the results of the models were compared. The proposed model also performed better than other architectures in terms of its specificity, sensitivity, and F-score, demonstrating a significant improvement in performance over the model. Because the proposed model was an ensemble model that employed XgBoost as a classifier and AlexNet as a feature extractor, it was superior to the original model. Additionally, the model we suggested was enhanced based on it. In the model, the AlexNet-extracted features were further sorted by the ReliefF algorithm, and the trial-and-error method was used to find the right number of feature inputs to improve the XgBoost classification performance and, as a result, the model’s overall accuracy. Table 7 also shows that model finished the classification task faster than the original model did (5.917 ± 0.001 s vs. 5.924 ± 0.001 s), while the accuracy went up from 95.145 ± 0.404% to 95.642 ± 0.398% (*p* = 0.05, *n* = 300). In terms of model correctness, the proposed model stood out among the models, with an accuracy rating of 95.642 ± 0.398%.

In general, the performance of the four compared models got better with each added element, proving unequivocally that under the suggested model, each component contributes favorably to the ensemble model’s performance improvement. The integrated model put out in this study can successfully and precisely carry out the identification and classification of eye-related diseases.

Table 7 presents the classification efficiency of the proposed deep-ocular model, which integrated AlexNet, ReliefF, and XgBoost, in comparison to various other architectures based on the total time taken in seconds for classification. It evaluates the performance using different feature extractors such as InceptionV3 and SqueezeNet combined with ReliefF and XgBoost. This table illustrates that the total time taken by the proposed model (AlexNet + ReliefF + XgBoost) was 5.917 s, with a very minimal standard deviation, and it was compared with other configurations in which the total times varied. Moreover, the performance of the proposed model with different feature selectors and final classifiers was evaluated. The models, including AlexNet combined with other algorithms like PSO, VGG16, Random Forest, and ELM, were compared in terms of the total classification time. The proposed model consistently showed a lower total time for classification in comparison with configurations like AlexNet + ReliefF + Random Forest and AlexNet + ReliefF + ELM, which took 7.271 s and 6.029 s, respectively, for total classification. The results emphasize the efficiency of the deep-ocular model in terms of classification time. Meanwhile, as shown in Table 7, there were differences in the time needed for different networks to extract features. In addition to accuracy, the AlexNet we used as feature extractor has the shortest running time (5.917 ± 0.001 s), which demonstrates the higher efficiency of our proposed model (AlexNet + ReliefF + XgBoost).

Figure 7 illustrates the confusion matrix derived from the proposed model for the detection of the three main classes of eye-related diseases, cataract, diabetic retinopathy and glaucoma, compared to normal retina. Additionally, as demonstrated in Table 7, the proposed model in this study performed significantly better than the current models in terms of efficiency, classifying the test set in just 5.917 ± 0.001 s (*n* = 300), thanks to its smaller neural network construction. Our model can drastically reduce training time while assuring accuracy. The AUC values for our model (given in Figure 8) improved generally, as can be observed. In comparison to previous methods, the ensemble model suggested in this work can generally identify eye diseases with more efficiency and accuracy.

### 4.3. Feature Visualization through AI Interpretability

Grad-CAM, which stands for gradient class activation map, is a technique used to visualize the regions of an image that a deep neural network focuses on when making a particular prediction. This visualization helps to provide insights into what parts of the input image contribute most to the network’s decision-making process. By highlighting these regions, Grad-CAM offers a better understanding of the features that the network is recognizing. To integrate Grad-CAM into a model like our deep-ocular system, a few steps are involved. First, you load your pre-trained model (in this case, using a modified AlexNet) and the image you want to visualize. The model should have a global average pooling layer followed by a dense layer for classification. Figure 9 shows the Grad-CAM visualization of features classified using the proposed deep-ocular system.

The process began by obtaining the index of the predicted class. Then, we calculated the gradients of the output score for the predicted class with respect to the final convolutional layer’s output. These gradients indicate the importance of different parts of the convolutional layer’s output in making the specific prediction. The higher the gradient, the more influential that part of the image is in predicting the chosen class. After obtaining the gradients, we calculated the heatmap by averaging the gradients along the channels of the convolutional layer’s output. This heatmap was then normalized and resized to match the dimensions of the original input image. The heatmap was further enhanced by applying a color map, often the “jet” color map. Finally, the heatmap was overlaid on the original image using an additive blending approach. The resulting image shows a visual representation of where the model’s attention was focused when making the prediction.

## 5. Discussion

Globally, the prevalence of eye problems varies greatly according to age, gender, occupation, lifestyle, economic status, hygiene, habits, and traditions. According to a study conducted on people in tropical and temperate areas, unavoidable eye infections are more common in low-altitude populations because of factors such as dust, humidity, sunlight, and other natural conditions [5]. Furthermore, ocular problems manifest differently in underdeveloped and developed cultures. Many underdeveloped nations, particularly in Asia, have true rates of ocular morbidity that are underdiagnosed and neglected [6]. A total of 285 million people are thought to have visual impairments worldwide, of which 246 million have poor vision and 39 million are blind [7]. According to the World Health Organization (WHO), 2.2 billion people worldwide suffer from nearsightedness or farsightedness [8]. According to estimates, 50% of these circumstances might have been prevented or resolved [9]. Cataracts, age-related macular degeneration, glaucoma, diabetic retinopathy, corneal opacity, trachoma, hypertension, and other disorders are among the main causes of visual impairment [10].

Deep learning-based algorithms are increasingly being used in medical-picture analysis [15]. Several tasks, including identifying illnesses, have shown the effectiveness of deep-learning-based models [16]. Automated illness diagnosis is one of the most crucial advances in reducing an ophthalmologist’s workload. Deep learning and computer vision techniques can be used to identify illnesses without the need for human input. Even though several of these studies have yielded hopeful findings, only a limited number of them have been able to fully diagnose more than one eye disease [17]. More research is needed to analyze the different components of fundus imaging in order to accurately detect a variety of eye disorders [18]. This study suggests using deep learning (DL) to identify various eye diseases.

Another tactic has been to use multilabel classification [19]. The databases on ocular diseases [37,38] are very uneven. Due to this imbalance, it is challenging to correctly detect or categorize disease or even a typical retinograph image. Due to its poor accuracy, this approach is not suggested for situations involving broad classification. The objective of this effort was to categorize ocular diseases. Given how unevenly distributed the dataset utilized in this study was, it is not advised to categorize any diseases using it. Because of this mismatch a lot of variety during training was undesirable. By balancing the representations of the classes, we were able to address this problem. We created a technique for deep-ocular detection using retinal fundus images. We made this system by using state-of-the-art image classification techniques, such as ensemble-based transfer learning, on the ODIR and RFMiD datasets, which have many images of different classes of the fundus. To diagnose eye-related disorders, the current models only use one deep learning network; therefore, the method’s effectiveness still has to be improved. In this study, we proposed a hybrid ensemble model with three steps that included a classifier, a feature extractor, and a feature selector. In order to avoid gradient vanishing and overfitting, the original picture features were first retrieved using a pre-trained AlexNet model with an improved structure. The most important features were then selected from the retrieved features using the ReliefF method in order to reduce the feature dimension. An XgBoost classifier gave classification results based on the selected features at the end.

The training–test split technique from the scikit-learn package was applied. A total of 70% of the data were used for training and 30% for testing, dividing the total into a 70:30 ratio. When the XgBoost classifier was fed the top 20 significant features, which was the highest value when compared to other numbers of input features, the accuracy of the classification results could reach 95.33%, as shown in Figure 7. Even though the accuracy can remain constant when only the first seven characteristics are entered, a larger number of features necessitates a longer training period for the model. Therefore, it was determined that the top five features offered by the AlexNet and ReliefF algorithms were the most appropriate for this application. All of the models had respectable accuracy, as seen in Table 4. The model had a structure that was superior to the original AlexNet. The results show that the accuracies of the two models were comparable, but the proposed model (improved AlexNet) may significantly help a model perform better, since if a model is built using one model rather than another model, it must classify 1000 features, significantly increasing the training time without clearly improving accuracy.

When comparing the performance of the proposed iAlexNet + ReliefF + XgBoost classifier to the studies listed in Table 1, several considerations were observed. The proposed system boasted an accuracy (Acc) of 95.642%, a sensitivity (SE) of 98.644%, and a specificity (SP) of 98.643%, with an F1 Score of 98.639 and an area under the curve (AUC) of 95. These metrics collectively offer a comprehensive insight into the model’s robustness and reliability. In terms of accuracy, the proposed system aligned competitively with several studies noted in Table 1. For instance, its accuracy was in the same range as studies [24,32], which reported accuracies around 96% and 95%, respectively. However, it did not surpass the accuracy of the methodology reported in [23], which achieved an impressive 99.11% accuracy. Nevertheless, the proposed model outperformed those in other studies like [26,27,28,29,30,31], whose accuracies fell below 95%. Evaluating on the basis of sensitivity and specificity, the proposed model demonstrates high performance, with both metrics exceeding 98%. This indicates a solid ability to correctly identify positive and negative cases, showcasing the model’s effectiveness.

The integrated module successfully and precisely identified and classified eye-related diseases. The proposed model (improved AlexNet, ReliefF, and XgBoost) had the highest accuracy value, demonstrating its excellence. The feature extraction component had the greatest impact on the model’s accuracy, with InceptionV3, SqueezeNet, and AlexNet being the most accurate. Finding a suitable network for feature extraction is crucial for successful classification. This study also compared the performance of five existing models with the new model to confirm its usefulness. The proposed model had the best overall accuracy rate (95.642%), reducing the training time and ensuring the accuracy, as reported in Table 6 and Figure 7. The ensemble model improves our overall efficiency and accuracy in identifying eye diseases compared to previous methods.

Utilizing the ODIR and RFMiD datasets, our approach integrated cutting-edge image classification techniques through an ensemble-based transfer learning approach. Central to our proposal was a novel three-step hybrid ensemble model, encompassing a classifier, feature extractor, and feature selector. Our approach harnessed the potential of an improved AlexNet architecture, featuring attention and dense layers, to enhance feature extraction, adaptability, interpretability, and accuracy. This architecture demonstrated superior performance compared to other transfer learning approaches, especially for retinograph classification tasks.

Moreover, our framework incorporates the ReliefF feature selection method to retain the most salient features and reduce dimensionality. The subsequent classification using XgBoost is capable of accurately identifying different ocular diseases. By integrating data augmentation techniques, we effectively address class-imbalance concerns. Notably, our deep-ocular model, predominantly built upon the AlexNet-ReliefF-XgBoost ensemble, achieved an impressive accuracy of 95.13%. This underscores the potential of our proposed approach to significantly aid dermatologists in making informed decisions for the early diagnosis and screening of eye-related diseases, ultimately contributing to improved patient care and outcomes.

### 5.1. Advantages of Proposed System

We developed an intelligent detection system for the early diagnosis of various eye-related diseases using retinal fundus images. This is a significant contribution, as automated systems can greatly aid clinicians and less-experienced ophthalmologists in identifying such diseases accurately and efficiently. The deep-ocular system’s approach incorporates deep learning (DL) algorithms to enhance classification results, which is a common and effective strategy for image-based medical diagnosis. The use of ensemble-based transfer learning (TL), combining a classifier, a feature extractor, and a feature selector, demonstrates a comprehensive approach to improving classification performance. The following are the key steps in the deep-ocular system:(1)We used the ODIR and RFMiD datasets, which provide a diverse set of retinal fundus images representing various classes, including glaucoma, diabetic retinopathy, cataract, and normal eyes.(2)We utilized a basic pre-trained AlexNet model and improved it through a self-attention mechanism and dense layers as an effective feature extractor. This model is known for its effectiveness in image classification tasks. Enhancements to the model’s structure were made to address overfitting and gradient vanishing issues. Extracted features from the images served as inputs for subsequent steps.(3)The ReliefF method was applied to select relevant features and reduce dimensionality. This is crucial for improving model efficiency and reducing the risk of overfitting.(4)An XgBoost classifier was employed for the final classification task. XgBoost is a popular machine learning algorithm that excels in tabular data and has been effective in various domains, including medical diagnosis.(5)To tackle class imbalance issues, we employed data augmentation techniques. This helped ensure that the model was exposed to a balanced representation of different classes, preventing it from being biased toward the majority class.(6)The proposed ensemble model, named “Deep-ocular,” achieved an impressive accuracy of 95.13%. This accuracy suggests that our system is highly capable of accurately classifying various ocular diseases.(7)The high accuracy achieved by our model indicates its potential to be a valuable tool for dermatologists and healthcare professionals, aiding them in making early and accurate decisions for diagnosing and screening eye-related diseases.

### 5.2. Limitations of Proposed System

This work addresses a critical need in the field of ophthalmology by providing an automated system that can assist medical professionals in identifying eye diseases more effectively. The integration of deep learning, ensemble methods, and data augmentation showcases a comprehensive approach to developing a robust diagnostic tool. However, there are some limitations of the proposed deep-ocular system, as described in Table 8.

## 6. Conclusions

In conclusion, the challenges faced by clinicians and less-experienced ophthalmologists in identifying early eye-related diseases have underscored the need for automated ocular disease detection systems. Manual diagnosis is riddled with labor-intensive efforts and the potential for errors, magnified by the diversity of ocular conditions such as glaucoma, diabetic retinopathy, cataract, and normal cases. In response, this study presents a comprehensive solution in the form of an intelligent detection system for retinal fundus images. Leveraging the power of deep learning algorithms, particularly the iAlexNet architecture enhanced with attention and dense layers, offers a paradigm shift in feature extraction, adaptability, interpretability, and accuracy within the realm of retinograph classification.

To realize this solution, this study harnessed the potential of ensemble-based transfer learning, establishing a three-step hybrid ensemble model that seamlessly integrated a classifier, a feature extractor, and a feature selector. The initial phase involved feature extraction utilizing a pre-trained AlexNet model, tailored with attention and dense layers to heighten its performance. This improved architecture, iAlexNet, offers distinct advantages over other transfer learning models, catering to the unique demands of retinograph classification tasks. The feature selection phase employs ReliefF to identify and retain relevant features, while the dimensionality is minimized for computational efficiency. Ultimately, a robust XgBoost classifier culminates in disease classification, yielding differentiated outcomes for various ocular conditions.

The efficacy of the proposed model was evidenced by the achievement of a remarkable 95.13% accuracy in the deep-ocular model. These promising results underscore the potential of this ensemble approach, affirming its viability as a valuable tool for aiding healthcare professionals in the timely diagnosis and screening of eye-related diseases. By combining cutting-edge techniques in deep learning, feature selection, and classification, this study contributes to a pivotal advancement in ocular disease detection and management.

## Figures and Tables

**Figure 1 diagnostics-13-03165-f001:**
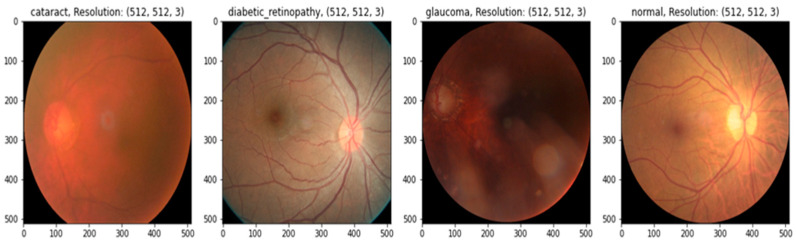
A visual example of ocular eye-related diseases studied.

**Figure 2 diagnostics-13-03165-f002:**
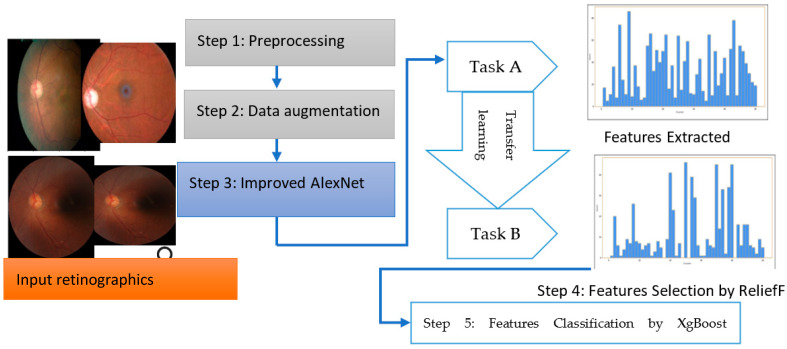
The proposed deep-ocular block diagram.

**Figure 3 diagnostics-13-03165-f003:**
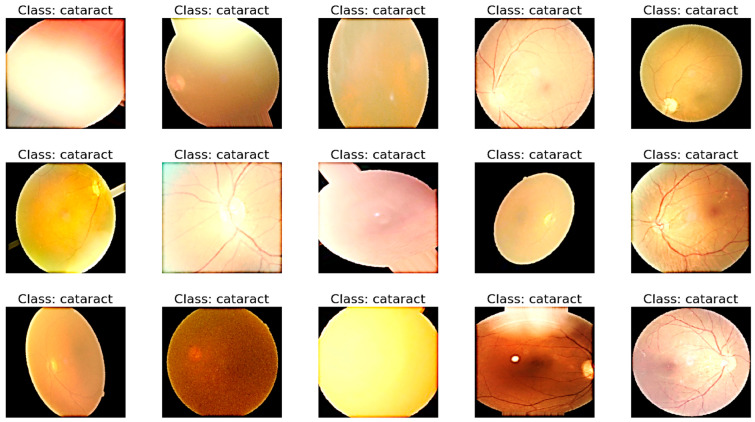
A visual example of preprocessing and data augmentation to control class imbalance.

**Figure 4 diagnostics-13-03165-f004:**
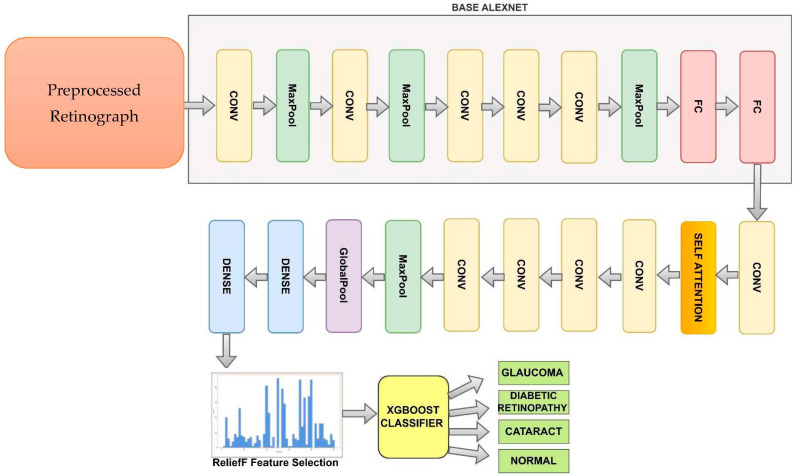
A modified AlexNet Architecture (iAlexNet) with Relief method and XgBoost classifier to recognize eye-related diseases.

**Figure 5 diagnostics-13-03165-f005:**
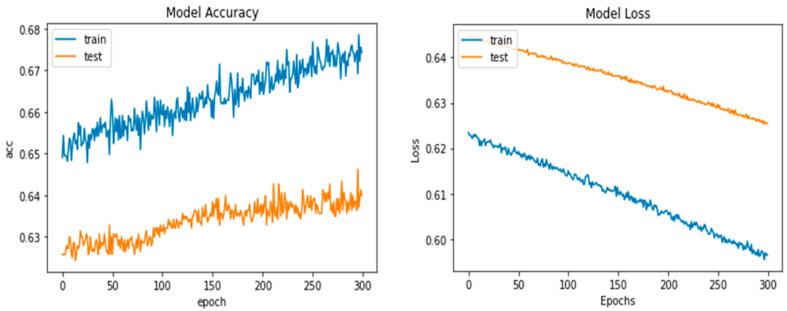
Proposed deep-ocular model accuracy and loss for both training and validation after 300 epochs.

**Figure 6 diagnostics-13-03165-f006:**
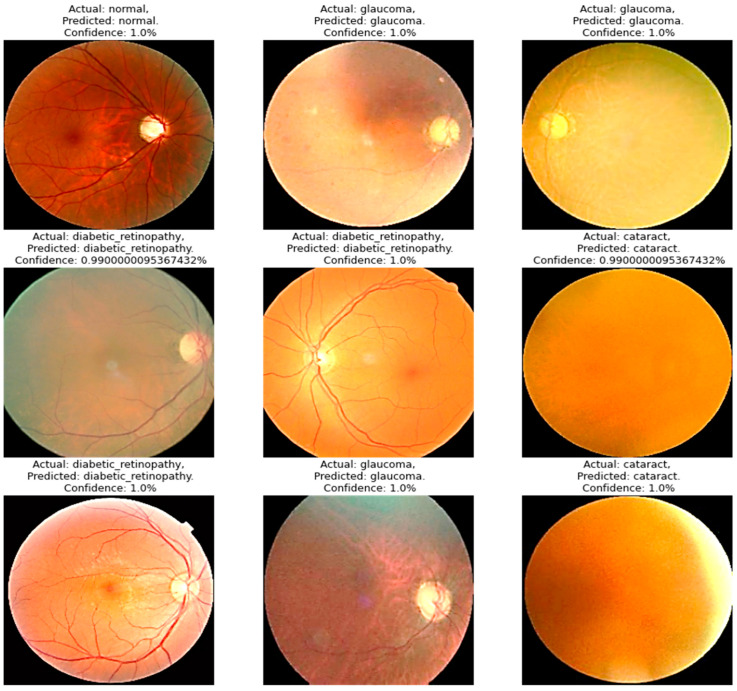
A visual diagram of results derived from the proposed model for the detection of four classes of eye-related diseases.

**Figure 7 diagnostics-13-03165-f007:**
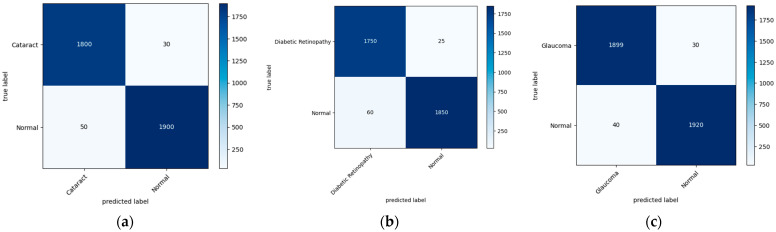
A visual diagram of the confusion matrix derived from the proposed model for the detection of four classes of eye-related diseases: (**a**) cataract, (**b**) diabetic retinopathy, and (**c**) glaucoma, against normal.

**Figure 8 diagnostics-13-03165-f008:**
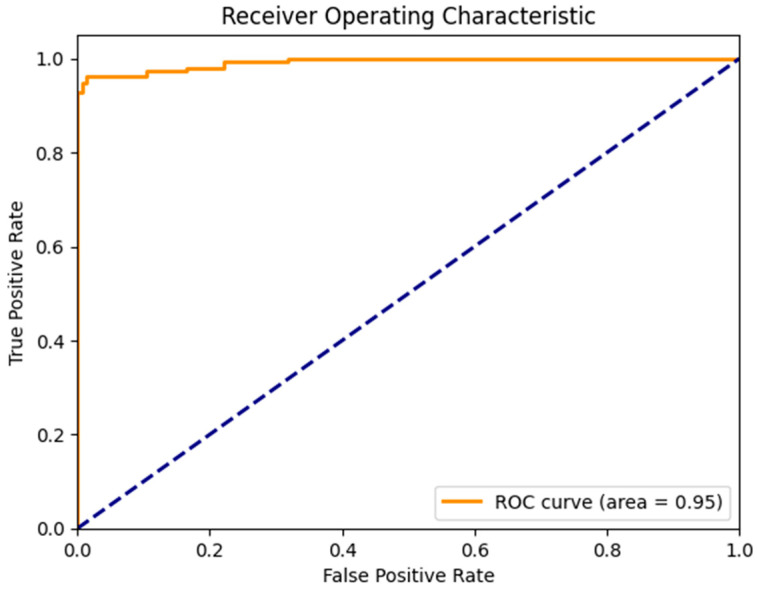
AUC of a proposed model to detect glaucoma and cataract eye-related disease.

**Figure 9 diagnostics-13-03165-f009:**
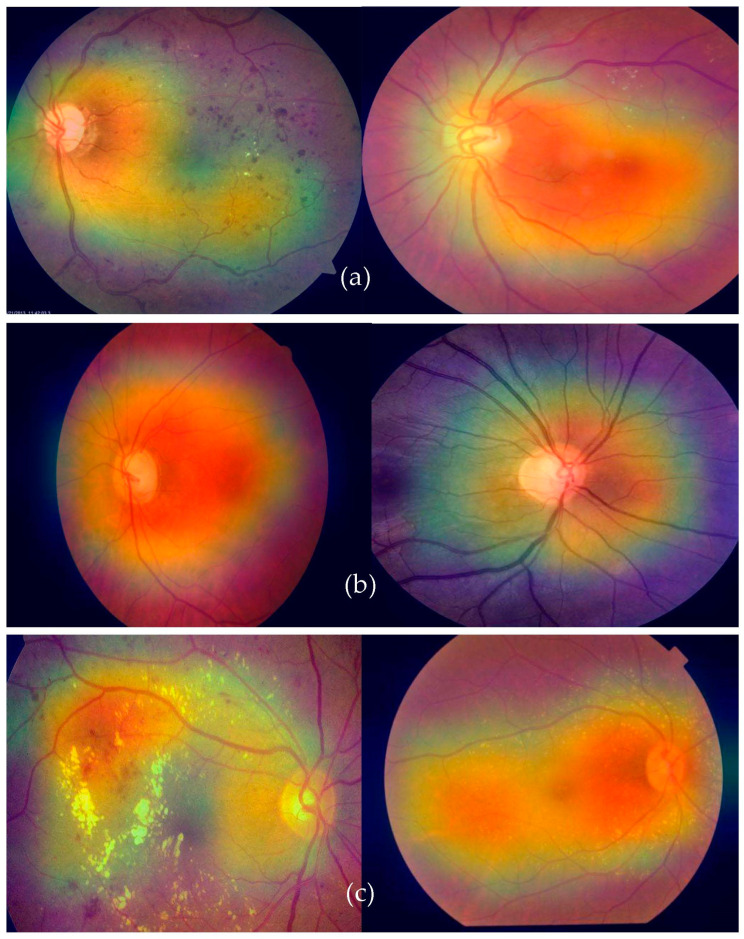
A visual Grad-CAM of the proposed deep-ocular system on retinographs, where (**a**) represents diabetes, (**b**) shows the glaucoma, and (**c**) indicates the cataract eye-related diseases.

**Table 1 diagnostics-13-03165-t001:** A comparative study of state-of-the-art techniques to recognize ocular diseases.

Ref.	Methodology	Results	Limitations
[23]	CNN for OD localization. Replacement of blue channel with segmented vasculature maps.	ACC: 99.11%	Limited eye diseases with huge computational power.
[24]	Hybrid DL network with ResNet50 for DR detection and grading.	ACC: 96%	Only detection of DR, and computationally expensive.
[25]	Single network for multiple binary predictions on DR severity.	ACC: 92%	Only detection of DR, and computationally expensive.
[26]	VGG19 model for cataract detection using color fundus images.	ACC: 91%	Need for expert ophthalmological intervention.
[27]	Use of comprehensive OIA-ODIR dataset with nine CNNs for multi-disease classification.	ACC: 90%	Computationally expensive and unbalanced and limited datasets.
[28]	CNN model for multi-class DED classification.	ACC: 81.33%	Limited dataset and no generalized solution for the detection of ocular diseases.
[29]	Optimal residual deep neural networks for diagnosing DR, glaucoma, and AMD.	ACC: <94%	Recognize limited categories of eye diseases and accuracy is not up to the mark.
[30]	VGG16 model for retinal complication diagnosis using retinal fundus images.	<94% of ACC	Detection of limited diseases and computationally expensive.
[31]	Deep learning for eye disease classification with minimal resolution images.	90%	Only detection of DR and computationally expensive.
[32]	Resnet152-based CNN for glaucoma detection.	95%	Detection of glaucoma only and computationally expensive.
[33]	Deep learning algorithm with AS-OCT images.	ACC < 93%	Only detection of DR and computationally expensive.
[34]	EyeDeep-Net, a multi-layer neural network for diagnosing various eye problems.	Significant superiority over baseline	Used unbalance dataset and model overfitting, and computationally expensive.
[35]	DCNN optimized with whale optimization algorithm for the detection of multiple eye diseases.	8.1% increase in accuracy	Used unbalance dataset and model overfitting.
[36]	DCNet for patient-level multi-label ocular disease classification with a multi-label soft margin loss.	Enhanced classification performance	Limited eye diseases with huge computational power.

**Table 2 diagnostics-13-03165-t002:** Distribution of original datasets obtained from ODIR and RFMiD datasets.

Data Set	Images *	Size	Type
1. ODIR	284 + 16 (PRV)	224 × 224	Glaucoma
	500	224 × 224	Diabetic retinopathy
	293 + 7 (PRV)	224 × 224	Cataract
	900	224 × 224	Normal
2. RFMiD	300	4288 × 2848 to 2144 × 1424	Glaucoma
	370 + 130 (ODIR)	4288 × 2848 to 2144 × 1424	Diabetic retinopathy
	300 (ODIR)	4288 × 2848 to 2144 × 1424	Cataract
	900	4288 × 2848 to 2144 × 1424	Normal

* PRV: Privately collected from different online sources, ODIR: Ocular Disease Intelligent Recognition, RFMiD: retinal fundus multi-disease image dataset.

**Table 3 diagnostics-13-03165-t003:** Data augmentation technique applied to selected datasets.

Augmentation Method		Parameters
Color Enhancement	Brightness change	[−25% to +25%]
	Contrast enhancement	CLAHE and Bengrams
Affine Transformation	Flipping	Vertical, horizontal, and both directions
	Rotation	−90°, +90°, −90°, +90°

**Table 4 diagnostics-13-03165-t004:** Parameters used in AlexNet training.

Parameters	Value
Model initial learning rate	5 × 10^−4^
Model learning rate and drop factor	0.1
Regularization term: L2 regularization	1 × 10^−4^
Max number of epochs	100
Mini batch size	32

**Table 5 diagnostics-13-03165-t005:** Classify ocular diseases by using different TL models.

**Original AlexNet (*n* = 400)**				
**Classification**	**Accuracy**	**Specificity**	**Sensitivity**	**F-Score**
1. Glaucoma	96.864 ± 1.655%	98.244 ± 1.713%	91.656 ± 6.102%	94.704 ± 3.078%
2. Diabetic Retinopathy	95.272 ± 1.835%	92.110 ± 4.613%	97.833 ± 2.290%	94.803 ± 2.389%
3. Cataract	93.528 ± 0.675%	98.145 ± 1.767%	97.611 ± 1.978%	97.854 ± 0.996%
4. Normal	96.528 ± 0.675%	98.145 ± 1.767%	97.611 ± 1.978%	97.854 ± 0.996%
Total	95.832 ± 1.895%	96.167 ± 1.540%	95.700 ± 2.007%	95.787 ± 1.953%
**Improved AlexNet (*n* = 300)**				
**Classification**	**Accuracy**	**Specificity**	**Sensitivity**	**F-score**
1. Glaucoma	97.897 ± 0.941%	98.980 ± 0.414%	94.233 ± 3.325%	96.516 ± 1.622%
2. Diabetic Retinopathy	97.648 ± 1.000%	94.656 ± 2.719%	98.833 ± 0.960%	96.676 ± 1.362%
3. Cataract	97.648 ± 1.000%	94.656 ± 2.719%	98.833 ± 0.960%	96.676 ± 1.362%
4. Normal	98.872 ± 0.229%	98.466 ± 1.208%	98.278 ± 1.155%	98.360 ± 0.329%
Total	97.208 ± 0.955%	97.367 ± 0.821%	97.115 ± 1.027%	97.184 ± 0.977%
**Improved AlexNet + XgBoost (*n* = 200)**				
**Classification**	**Accuracy**	**Specificity**	**Sensitivity**	**F-score**
1. Glaucoma	98.834 ± 0.436%	98.279 ± 0.555%	97.975 ± 1.262%	98.123 ± 0.712%
2. Diabetic Retinopathy	98.642 ± 0.343%	97.289 ± 1.350%	98.833 ± 0.665%	98.047 ± 0.478%
3. Cataract	98.642 ± 0.343%	97.289 ± 1.350%	98.833 ± 0.665%	98.047 ± 0.478%
4. Normal	98.815 ± 0.217%	98.934 ± 0.546%	97.611 ± 0.644%	98.266 ± 0.320%
Total	98.145 ± 0.404%	98.168 ± 0.384%	98.40 ± 0.426%	98.145 ± 0.410%
**The proposed model (Improved AlexNet + ReliefF + XgBoost) (*n* = 300)**				
**Classification**	**Accuracy**	**Specificity**	**Sensitivity**	**F-score**
1. Glaucoma	94.082 ± 0.335%	98.412 ± 0.577%	98.650 ± 0.950%	98.528 ± 0.540%
2. Diabetic Retinopathy	95.082 ± 0.369%	98.302 ± 1.061%	99.056 ± 0.375%	98.674 ± 0.528%
3. Cataract	93.120 ± 0.273%	99.218 ± 0.467%	98.222 ± 0.861%	98.715 ± 0.403%
4. Normal	96.642 ± 0.398%	98.644 ± 0.388%	98.643 ± 0.406%	98.639 ± 0.398%
Total	95.642 ± 0.398%	98.644 ± 0.388%	98.643 ± 0.406%	98.639 ± 0.398%

**Table 6 diagnostics-13-03165-t006:** Classification time consumed by models iAlexNet + XgBoost and iAlexNet + ReliefF + XgBoost (*n* = 300).

Models	Feature Extraction Time/s	Time for Classifier/s	Total/s
iAlexNet + XgBoost	5.896 ± 0.001	0.028 ± 0.000	5.924 ± 0.001
iAlexNet + ReliefF + XgBoost	5.896 ± 0.001	0.022 ± 0.001	5.917 ± 0.001

**Table 7 diagnostics-13-03165-t007:** Classification efficiency of proposed deep-ocular (our proposed model (AlexNet + ReliefF + XgBoost)) model compared to other different architectures in terms of time in seconds.

**Using Different Feature Extractors**	
**Models**	**Total Time/s ***
AlexNet + ReliefF + XGBOOST	5.917 ± 0.001
InceptionV3 + ReliefF + XGBOOST	6.428 ± 0.001
SqueezeNet + ReliefF + XGBOOST	5.918 ± 0.002
**Using Different Feature Selectors**	
AlexNet + ReliefF + XGBOOST	5.917 ± 0.001
**Models**	**Total time/s**
AlexNet + PSO + XGBOOST	5.918 ± 0.001
AlexNet + VGG16 + XGBOOST	5.918 ± 0.001
AlexNet + ReliefF + XGBOOST	5.917 ± 0.001
**Using Different Final Classifiers**	
**Models**	**Total time/s**
AlexNet + ReliefF + XGBOOST	5.917 ± 0.001
AlexNet + ReliefF + Random Forest	7.271 ± 0.001
AlexNet + ReliefF + ELM	6.029 ± 0.001

* The time required for total classification in seconds.

**Table 8 diagnostics-13-03165-t008:** Limitations of proposed system, when further developing and deploying the deep-ocular system in clinical or research settings.

Limitation	Description
Limited Generalization	The system’s performance might degrade when applied to datasets from different sources or populations due to differences in imaging quality, patient demographics, and diseases. In addition, other ocular diseases such as hypertension should be considered.
Data Availability	The system’s performance heavily relies on the availability and diversity of high-quality labeled retinal fundus images for training. Limited data could hinder its effectiveness.
Overfitting	Despite enhancements, overfitting might still occur due to the complexity of deep learning models. Regularization techniques might be needed to mitigate this issue.
Limited to fundus images	The system’s applicability is limited to eye diseases that can be diagnosed using retinal fundus images. Conditions that require other types of tests will not be detected.
Computational Resources	Training and utilizing deep learning models can demand significant computational resources, which might limit the system’s accessibility in resource-constrained settings.

## Data Availability

The Python code is provided on GitHub (https://github.com/Qaisar256/Deep-ocular) to better understand the deep-ocular system.

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
