# Peer review of "Deep-Ocular: Improved Transfer Learning Architecture Using Self-Attention and Dense Layers for Recognition of Ocular Diseases"

_diagnostics, 2023, doi:10.3390/diagnostics13203165_

Round 1

Reviewer 1 Report

An intelligent detection system based on recurrent fund images has been developed in this article. The above research has practical value for the diagnosis of early eye diseases. The content of this article is complete and experimental verification is reasonable. It is recommended to make minor revisions before publication. Suggest the author to improve the quality of the image, as there may be some horizontal and vertical imbalance, such as Figure 8 and Figure 9.

Author Response

Original Manuscript ID:  ID: diagnostics-2607800       

Original Article Title: Deep-Ocular: Improved Transfer Learning Architecture by Self-Attention and Dense Layers for Recognition of Ocular Diseases

To: Editor in Chief,

MDPI, Diagnostics

Re: Response to reviewers

Dear Editor,

Many thanks for insightful comments and suggestions of the referees. Thank you for allowing a resubmission of our manuscript, with an opportunity to address the reviewers’ comments.

We are uploading (a) our point-by-point response to the comments (below) (response to reviewers), (b) an updated manuscript with green, blue, and orange highlighting indicating changes, and (c) a clean updated manuscript without highlights (PDF main document).

By following reviewers’ comments, we made substantial modifications in our paper to improve its clarity, English and readability. In our revised paper, we represent the improved manuscript such as:

(1) Revised Abstract, (2) Revised Introduction, (3) Results section, (4) Discussions and Conclusion sections.

We have made the following modifications as desired by the reviewers:

Best regards,

Corresponding Author,

Dr. Qaisar Abbas (On behalf of authors),

Professor.

Reviewer 2 Report

Overall the paper presents rather interesting results, which involve combination of transfer learning of convolutional neural networks, feature selection as well as XgBoost for final classification

However reviewer noticed several issues that does not allow to publish paper in its current form.

One part of questions is related to the methodology and another related to presentation.

Regarding methodology.

From the table 5 and supporting comments it is not clear why the proposed model is better: it has lower accuracy , at cost of higher specificity and sensitivity. The increse of specificity and sensitivity is about 0.5-1.5 %, while accuracy can drop up to 4%. without additional comments it looks like the proposed model "feels" weaker.

In table 8 authors provide an overview of the different network models

But from it and supporting comments it is not clear why AlexNet is go-to network, as it has way more parameters than versions of inception, more operations (and longer inference time), as vell as lower performance on ImageNet that all of other architectures. Presented advantages (from 1013 - 1036) conteins either general information that is not specific to the AlexNet or is simply not true ("simple architecture and computational efficiency" at line 1031)

Additionaly discussion would benefit from referencing to limitations in table 1 and how proposed work challenges such limitations

The parper is hard to comprehense, due to its big size and overload of the relatevely generally known information:

* abundant description of alexnet, which consist of 20-lines of architectural description (425-446), as well as additional image description, on top of paragrapgh of general description

*description of generallly known XgBoost classifier also takes a lot of space - lines 540-576

Additionally the list of references requires revision as the referenced article 24 in the paragrapghs about XgBoost actually is in no way related to the XgBoost algorithm, in cited paper authors use only CNN   C. S. Lee, A. J. Tyring, N. P. Deruyter, Y. Wu, A. Rokem, and A. Y. Lee, “Deep-learning based, automated segmentation 1220 of macular edema in optical coherence tomography,” Biomedical Optics Express, vol. 8, no. 7, pp. 3440–3448, 2017.

Possible other references are also misplaced 

Paper is defenetely required another round of proof reading, as there are some repetitions of more or less same text, example of it are lines 228-233, and lines 255-260 looks far too similar , or lines 390-392 and 403-404 convay more or less the same information, but repeated. Same could be said about paragrahs at lines 1014-1024 and 1025-1035

Lines 686 to 697 has a feeling as being written in either poor English, or by ChatGPT-like engine, as has weird wording. Expecially what is concerning is the mention of RDF Kernels in context of XGBOOST (which is weirdly written as XgBoost and XGBOOST in a single paragraph).

In general there is such impression that some paragrpahs were written by ChatGPT-like engine. (Or it could be just bad english writing)

There are several typos in folmulas (like in formula 1 line 514 the is sum over j, but no j index in the summed variables)

In table 5, in part about Improved AlexNet + XGBOOST (n = 200) the total accuracy is miscalculated 9.145 - probably a typo (right before lone 709)

Algorithms requires revision. Algorithms 1, 2 and 4 could be rewritten as there is simply too much text. Algorithm 3 will benefit from indentations for a better comprehension. Also in algorithm 4 is the only time where the residual blocks are mentioned 

Figures could benifit a revision too:

* figure 3 could be removed all together as it does not add any useful insights 

* figure 6 shows that actually the epochs number could still be increased, as there is clear linear trend

* figure 7. Authors present condifence in % format, however the values are in range from o to 1

* figure 8 has two weirdly stacked confusion matrices 

* AUC curve at figure 10 seems to have too low of AUC value (as apposed to the high accuracies, sensisivities etc)

* figure 11 classification errors could be removed as it provides no useful information (raw numbers without total values)

Not sure if this is relevant, by some tables and figures have formatting errors (line numbers overlap with the content)

f.e. line 305 -  316, 328 - 336, 524- 538, 643 - 651, 827 - 835

Author Response

(The authors gave the same response as above.)

Round 2

Reviewer 2 Report

The authors seems to rather adequately reacted to the comments of the previous review. In particular I appreciate revision of the text as well as reconsideration of including/excluding information in forms of figures and tables. 

Text feels fresher and less repetitive. 

Some remaining issues in the paper:

* after suggested removal of figures not all titles and references are corrected (there is reference to figure 9 that is missing at line 906 / figure 10 at line 942 is mislabled, at line 1038 you reference table 7 for accuracy, though in tables 7 the computation time is implies). Please revisit this moment.

* algorithms could still benefit from adding extra indentation while showing the loops, the 2nd algorithm is hard to comprehend due to long names of functions

* figure 7 was created with mistakes as ideally with high reported values confusion matrix should be diagonal 

* formatting issues with references to figures (Figure (with capital F) or figure, and Fig.), tables (table or Table), XgBoost (there is all capitals XGBOOST and XgBoost)

Author Response

Dear Reviewer,

We have revised the paper as requested by you.

Thank you very much.
